# Comparison of Road Noise Policies across Australia, Europe, and North America

**DOI:** 10.3390/ijerph19010173

**Published:** 2021-12-24

**Authors:** Maxime Perna, Thomas Padois, Christopher Trudeau, Edda Bild, Josée Laplace, Thomas Dupont, Catherine Guastavino

**Affiliations:** 1Department of Mechanical Engineering, École de Technologie Supérieure (ÉTS), Montreal, QC H3C 1K3, Canada; thomas.padois@etsmtl.ca; 2School of Information Studies, McGill University, Montreal, QC H3A 1X1, Canada; christopher.trudeau@mail.mcgill.ca (C.T.); edda.bild@mcgill.ca (E.B.); josee.laplace@mail.mcgill.ca (J.L.); 3Centre for Interdisciplinary Research in Music Media and Technology, School of Information Studies (CIRMMT), McGill University, Montreal, QC H3A 1E3, Canada

**Keywords:** environmental noise, road noise limits, road noise policies, acoustic measurement protocols

## Abstract

Developing innovative noise policies that build on international best practices is difficult when policies around the world differ along many dimensions, ranging from different sources covered to different levels of governance involved. This is particularly critical in the context of road traffic, identified as one of the main culprits leading to noise-associated complaints and health issues. In this article, we document the wide range of specifications observed in road traffic policies and propose a methodology to compare noise limits across noise policies. First, we present the responsibilities of administrative governments according to the scope (e.g., emission vs. exposure). Second, we compare noise limits by scope and geographic areas by separating acoustic indicators (overall and event indicators). Third, we convert overall outdoor noise limits into a common basis using the method described by Brink and his associates (2018) and compare them with the World Health Organization (WHO)’s recommendations (2018). Finally, measurement protocols are also compared across outdoor noise policies. This paper shows that road noise is managed at several administrative levels using approaches that are either centralized or decentralized. We also observed disparities in the associated noise limits across geographic areas. The converted outdoor noise limits generally exceeded the WHO’s recommendations (2018). Finally, this paper outlines how outdoor measurement protocols vary across geographic areas. However, similarities were identified between state and provincial noise policies within the same country.

## 1. Introduction

Environmental noise is a pollutant affecting health and well-being. It is defined by the World Health Organization (WHO) as the noise from all sources except occupational noise [1]. According to the WHO recommendations, long-term noise exposure at high levels can have detrimental effects on health, ranging from annoyance to sleep disturbance at nighttime, cardiovascular diseases and cognitive impairment in children [1]. Among the sources contributing to environmental noise, road noise is the one with the most serious and documented effect on large parts of populations around the world including the United States of America (USA), Australia, and most countries in Europe [2,3,4,5].

To regulate road noise, regulations, directives, and guidelines are set at different levels of administrative government of a geographic area/country. Depending on the country or the geographic area, noise policies may be set either at the highest administrative government (e.g., supranational or national) using a centralized approach or at a provincial or state level using a decentralized approach. To better address the different approaches, defining the noise limits in noise policies, several reviews of such policies have been conducted proposing noise limit comparison strategies [4,6,7]. The main issue remains that values in dB(A) are difficult to compare if they are not defined or measured using the same acoustic indicator.

To address this gap, we proposed a comparison strategy to compare noise limits, from Australia, Europe, and North America (the selection criteria are presented in Section 4.1 and the selected countries in Section 4.2), according to the administrative level of governance (where noise limits are defined), acoustic indicators (e.g., overall and event indicators), measurement protocols (e.g., acoustic standard, calibration, measurement location, weather conditions, and measurement periods), scopes (noise limits set for emission noise, exhaust, engine and tire noise or at outdoor and indoor premises exposed to road noise), noise exposure zones (sensitive, residential, mixed and industrial), and type of road (existing, upgrade and new road). The outdoor noise limits are also compared with the WHO’s recommendations [1] by using the conversion method of Brink et al. [8] (hereinafter referred to as “Brink’s method”) to convert noise limit values to the same acoustic indicators. The aim of this review was to address: (i) the influence of centralized and decentralized approaches on noise policy definition and (ii) to compare noise limits across countries.

The paper is divided into five parts: we first define the main acoustic indicators identified in noise policies. Second, the literature gaps are presented in literature review section. In the methods section, we present the selection criteria for noise policies analyzed in this study as well as Brink’s method used to compare noise limits with the latest WHO recommendations. The results section then provides (i) an overview of the scopes at different levels of governments; (ii) a descriptive analysis of the noise limits along relevant dimensions (i.e., by scope, geographic area, acoustic indicators and both, before and after conversion using Brink’s method for outdoor noise limits in comparison with the WHO’s recommendations); (iii) the associated measurement protocols. Finally, the discussion highlights the relative advantages of the centralized and decentralized approaches and compares the findings with the WHO’s recommendations in terms of acoustic indicators, measurement periods, noise limits, zones, and measurement protocols.

## 2. Definitions of Acoustic Indicators

Noise limits are defined using two types of acoustic indicators: overall indicators, used to measure the equivalent and averaged noise level over a period of time, and event indicators, used to represent the maximum noise level over a period of time (see Table 1). The overall indicators can refer to different measurement periods of a day (day, evening, night) or a full day with the use of two or three acoustic indicators (e.g., *L_d_*, *L_e_*, and *L_n_*). Otherwise, noise limits can be set for a full day with the use of a single indicator (e.g., *L_Aeq_*_,*24h*_, *L_dn_*, and *L_den_*).

These indicators are all based on the A-weighting, dB(A), considering the loudness perceived by the human ear for noise level lower than 55 dB. The indicators identified in noise policies are listed and defined in Table 1.

## 3. Literature Review

Four previous studies conducted reviews on noise policies by gathering noise limits [7] or by comparing the noise limits they included either by converting them to a common indicator [4] or without converting them at all [6]. The issue remains that values in dB(A) are difficult to compare if they are not defined or measured using the same acoustic indicator. In 2009, for instance, a noise policy review was conducted by the I-INCE (International Institute of Noise Control Engineering) Member Society to provide useful information for the development of worldwide noise policies to fight against environmental noise sources. A comparison strategy was proposed to gather noise limits according to the country, noise source, legal authority of the document, nature (emission or exposure), scope (free field, outdoor, and indoor), time of the day, measurement period, and acoustic indicators. However, noise limits were not compared. In 2016, outdoor noise limits from Australian policies at the state level using *L_A10%_* were converted into *L_Aeq_* using the approximation *L_Aeq_*_,T_ = *L_A10%_*_,18h_ − 3 dB(A) [4]. Burgess and Macpherson [4] showed that the noise limits adopted by the Australian states using *L_A10%_*_,18h_ were about 5 dB higher than those based on *L_Aeq_*_,T_ indicators. Considering this approximation as valid (*L_Aeq_*_,T_ = *L_A10%_*_,18h_ − 3 dB(A)), they concluded that there were disparities on the defined noise limits between the Australian state policies. In 2019, on the other hand, the European Network of the Heads of Environment Protection Agencies published an overview of noise limits from national policies [6]. This report showed that the majority of European national noise limits exceed the 2018 values recommended by the WHO [1], by comparing without converting outdoor noise limits based on *L_d_* (including the evening period) and *L_den_* [6]. In 2019, Brink et al. [8] proposed a method to convert noise values measured and calculated using *L_d_*, *L_e_*, *L_n_*, *L_dn_*, and *L_den_* into a common ground.

In 2005, Nijland and Van Wee [9] conducted a review on acoustic indicators and measurement protocols from road and rail noise policies from European countries. They showed that acoustic indicators and measurement protocols (e.g., measurement periods, time of the day, zones, and type of road) differ between policies. In this fact, a harmonization could make noise situations comparable among countries [9]. In the same way, in 2015, D’Alessandro and Schiavoni [10] reviewed the most used European noise map indices (calculated from acoustic indicators) for prioritizing mitigation measures in zones most affected by noise, and they concluded with the need of a common approach for a unified noise management strategy across member states of the European Union (EU) in order to improve noise mitigation efforts. The European Noise Directive (END) [11], aimed already back in 2007 at establishing a more centralized/uniformed approach by requesting the production of noise maps (CNOSSOS-EU (European Commission developed Common Noise Assessment Methods for Road, Railway, Aircraft, and Industrial Noise) [12]) and action plans with the aims of reducing environmental noise, but the END has not yet been fully implemented by the Member States [10]. Additionally, in 2017, Miloradović et al. identified regulations (in Europe) and acoustic standards (International Standard Organization) used to measure road noise emission [13]. However, to our knowledge, no review has been proposed that focuses on the actual measurement protocols (e.g., acoustic standard, calibration, measurement location, and weather conditions) used to ensure compliance with noise as part of policies for either indoor or outdoor environments.

## 4. Methods

### 4.1. Criteria for Selecting Noise Policies

This paper is based on an overview of 37 noise policies from Australia, Europe, and North America. These geographic areas were chosen for their similarities to the Quebec and Canadian contexts, and because their populations are heavily impacted by road noise.

Already in 1981, approximately 100 million people were exposed to road traffic noise in the USA [3]. In 2016, various studies undertaken by national and state environmental agencies in Australia have shown that the highest level of noise annoyance perceived indoors is associated with road traffic noise [4]. In 2017, noise monitoring surveys showed that road traffic is the main source responsible for high noise levels in low and middle income countries [5]. Finally, Europeans are more exposed to road traffic noise than other environmental noise sources such as rail, aircraft, or industry [2]. In 2020, at least 20% of the European population lived in zones where noise levels from road traffic are harmful to health [2].

To provide information relevant to the Quebec context, we selected Western countries that had noise policies and met at least one of the following criteria: (i) similarity of climatic conditions with Quebec, (e.g., cold winter, hot and humid summer); (ii) similarity with Quebec’s demographics (e.g., presence of large zones with low density and towns with high density); (iii) similarity in terms of governance. The noise limits from policies were obtained from the official websites of supranational/national and provincial/state ministries (e.g., environmental protection agencies) as well as legislative portals (which gather all laws in force and their up-to-date amendments). The keywords used for the search were translated into the different languages represented (e.g., “Lärm” in German and “ruido” in Spanish). The noise source (e.g., “road” OR “traffic” in English) was used to refine the search. Additional keywords were used to locate relevant information (e.g., “dB” for noise limits) in each document.

### 4.2. Methodology for Categorizing Noise Limits and Measurement Protocols

To compare the noise limits included in 37 noise policies, we first categorized them according to the level of administrative government at which they were set, the scope, the type of acoustic indicator, measurement period, and noise exposure zone.

Some indicators provide the overall noise level, while others measure events, emerging from ambient noise such as maximum/peak noise level. In order to compare noise limits, acoustic indicators were categorized into two categories: overall indicators and event indicators. Overall indicators, respecting the conversion criteria of Brink’s method (presented in Section 4.4), were also included.

The definition and number of zones associated to noise limits may also vary across noise policies. To allow for a comparison, the zones in the examined noise policies were mapped to the four categories used in Quebec’s 98-01 instruction note [14] (see Table 2).

The WHO’s noise guidelines provide recommendations for residential zones, all zones where dwellings occur. For our analysis, in cases where the zones were labeled as mixed-uses (including residences) or not specifically labeled, we considered them as residential zones.

In cases where measurement protocols (used to check the compliance with noise limits) were provided by or referred to in noise policies, we categorized the information according to the measurement location (e.g., microphone height, and minimum distance to reflective surfaces), weather conditions (e.g., wind speed, rainfall, and temperature), and sound level meter (e.g., type and calibration). The acoustic standards mentioned in noise policies were also collected.

### 4.3. Identified Noise Limits by Scope for Different Administrative Governments

A total of 522 noise limits were identified in the 37 noise policies across three geographic areas (Australia, Europe, and North America) and for various administrative governments—see Table 3.

Some countries/geographic areas use a centralized approach (e.g., countries in the EU and outside, for example, Norway) where noise limits are set by the highest administrative government (e.g., supranational and national levels), while other geographic areas/countries set noise limits at provincial or state levels (e.g., Australia and North America). This is why the noise policies selected here span the highest levels of administrative government.

The 522 noise limits include:287 noise limits defined for emission, i.e., noise levels emitted by vehicle engines, exhaust, or tires;192 noise limits defined outdoors, i.e., at the facade of noise-sensitive premises;43 noise limits defined indoors, of noise-sensitive premises.

Of these 192 noise limits defined outdoors, 111 were compatible with Brink’s method (criteria presented in Section 4.4), meaning that they were converted on a common ground for the purpose of comparison.

### 4.4. Methodology for Converting Outdoor Noise Limits

Brink et al. [6] proposed a method for converting noise exposure values, with associated uncertainties, toward common acoustic indicators according to the measurement periods used in the definition of *L_d_*, *L_e_*, *L_n_*, *L_Aeq_*_,24h_, *L_dn_*, and *L_den_* acoustic indicators, even when the nature and the circumstances of the measurement or calculation of these values are unknown. Due to the wide range of acoustic indicators identified in noise policies and considering that noise limits values are defined for measurements and calculations (when it comes to verifying their compliance), we assumed that Brink’s method is applicable for providing a common ground for comparison across noise limits identified for outdoors in order to compare them with the WHO’s recommendations (defined also for outdoors). However, this method can only convert noise limits based on overall indicators. The event indicators were therefore not converted, because no conversion method was applicable.

Measurement periods are defined differently across countries, states, and provinces due to the fact of local adaptions (see Section 5.3.1); hence, Brink’s method accounted for the measurement period of the original acoustic indicator. Table 4 shows the applicable measurement periods for Brink’s method. All conforming noise limits were converted to *L_den_* and *L_n_* indicators, defined for the following periods: day (*L_d_*) between 7 and 19 h, evening (*L_e_*) between 19 and 23 h, and night (*L_n_*) between 23 and 7 h (*L_den_* defined on these periods). These indicators and measurement periods were selected because they were used in the WHO’s noise recommendations, thereby allowing for a comparison with these.

It is important to stress that Brink’s method is applicable where vehicle types and driving behaviors are similar to Western European countries. Indeed, this method is based on the analysis of the diurnal traffic variation data collected in several Western Europe countries and the arithmetic difference between calculated *L_d_*, *L_e_*, *L_n_*, *L_Aeq_*_,24h_, *L_dn_*, and *L_den_* indicators over 50,000 dwelling facades in Switzerland (covering the full population of Swiss buildings) [8]. We assumed that this method is relevant for the selected western geographic areas, Australia, Europe, and North America presented in Table 3.

## 5. Results and Analysis

In this section, an overview of the responsibilities of governments at different levels and by scope is presented for the selected geographic areas. Then, all road noise limits identified in noise policies were compared by geographic area, scope, and type of acoustic indicators. Afterwards, the permitted outdoor noise limits to Brink’s method were converted and compared to the WHO’s recommendations. Finally, the outdoor measurement protocols are presented to know how compliance with noise limits is checked.

### 5.1. Scope at Different Levels of Government

#### 5.1.1. Emission Scope

Over the 287 noise limits identified for emission, 116 are defined for exhaust noise, 150 are defined for engine noise, and 21 are defined for tire noise. Noise limits for exhaust and engine noise are set by the highest administrative government, supranational (e.g., EU [23,24]) or national (e.g., Australia [15,52], USA [45], and Canada [38]). The EU set progressively more restrictive noise limits over a ten-year period, with noise limits entering into effect in 2016, 2022, and 2026 [24]. These noise emissions are presented by the manufacturer at the point of sale and must comply with the noise limits provided in European regulation [24]. States and provinces also had regulations, but covering emission values for road vehicles (e.g., Illinois [49], Washington [51], Quebec [43], British Columbia [41], and Western Australia [19]). In Quebec, for instance, noise limits may be checked on vehicles in circulation by a peace officer in order to determine if the owner of the vehicle can drive or allow his vehicle to be driven according to the measured noise level [43]. For tire noise, 21 noise limits are defined in EU [25]. The noise limits identified for emission (e.g., engine, exhaust, and tire noise) are defined according to various parameters presented in the Appendix A (see Table A1).

#### 5.1.2. Indoors and Outdoors

For indoors and outdoors (respectively, 43 and 192 noise limits), the noise limits are mainly defined by:National/federal governments for European countries;Provincial/state governments in Australia and North America;Zones, e.g., sensitive, residential, mixed, and industrial zones;Type of road, e.g., existing, planning new/upgraded roads.

Municipalities generally refer to higher administrative levels to regulate road noise for indoors and outdoors (e.g., Ontario [44] for Ottawa [53], Illinois [48] for Chicago [54], Switzerland [37] for the Canton of Geneva [55]).

#### 5.1.3. Noise Maps and Action Plans

The European Noise Directive (END) [11] aims to establish a centralized approach for the production of noise maps in order to target the zones exposed to noise. To fight against noise in the most exposed zones, action plans are undertaken to reduce environmental noise. For road noise, noise maps and action plans are set-up in large cities and agglomerations (population ≥ 100,000) and major road axes (transit ≥ 6 million per year). The END is incorporated into the regulations of the countries (e.g., France [56,57] and Germany [58]). In France [56,57], the action plans and noise maps are developed at different levels of administrative government:Departmental prefectures for municipalities with less than 100,000 inhabitants;Metropolises and municipalities greater than 100,000 inhabitants;Agglomerations (municipalities on the outskirts of large cities).

Agglomerations can develop a partnership with large cities’ noise observatories (e.g., Acoucité in Lyon [59] and BruitParif in Paris [60]) to produce noise mappings and action plans.

### 5.2. Noise Limits Summarized by Geographic Area, Scope, and Type of Acoustic Indicators

The type of acoustic indicators used by noise policies and regulations are presented in Table 5. The *L_Aeq_* and *L_Amax_* are the most identified indicators in noise policies, respectively, for overall and event indicators. For outdoor and indoor noise limits, the most frequently used indicator is *L_Aeq_*. However, some countries or states/provinces use *L_den_*, *L_d_*, *L_e_*, *L_n_*, and *L_dn_* indicators (see Table 5). Moreover, some adjustments based on average daytime and nighttime road traffic are applied to *L_Aeq_*; for example, Switzerland uses the acoustic rating indicator *L_Ar_* [37]. For emission scope, the main event indicator used is *L_Amax_* (except for Illinois [48,49], where *L_Aeq_* is chosen). In addition, the *L_Amax_* event indicator may be also used for outdoors, during daytime and evening (e.g., Sweden [35]). The *L_A5_*_%_ and *L_A10_*_%_ event indicators are also used, respectively, to characterize the impact of road noise during rush hour (e.g., Queensland [16]) and at nighttime (e.g., Norway [36]).

#### 5.2.1. Overall Indicators by Scope and Geographic Area

All noise limits based on overall indicators are gathered by geographic area and presented by emission, outdoor, and indoors, respectively, in Figure 1, Figure 2 and Figure 3. The dot plots show the noise limits for each scope, grouped according to geographic area. Each dot represents the value of a single noise limit; the dotted lines illustrate the minimum, median, and maximum values; *n* indicates the number of noise limits represented in each figure.

Overall indicators (e.g., *L_Aeq_*) are most often used for outdoor (*n* = 177) and indoor scopes (*n* = 42) as compared to the emission scope (*n* = 16), which is limited to the case of Illinois [48,49]. Indeed, there are no noise limits for the emission scope based on overall indicators identified in Australia and Europe (see Figure 1). For outdoors (see Figure 2), the median value was lower in Australia (55 dB(A)) than those presented in Europe (58 dB(A)) and North America (70 dB(A)), and the values were much less spread in Australia (max – min = 8 dB(A)) than those in Europe (max – min = 35 dB(A)) and North America (max – min = 50 dB(A)). Europe provides, however, the lowest median values (35 dB(A)) for indoors (see Figure 3), compared to those presented in Australia (40 dB(A)) and North America (45 dB(A)). Moreover, the values were also much less spread in Australia (max – min = 5 dB(A)) than those in Europe (max – min = 30 dB(A)) and North America (max – min = 20 dB(A)).

#### 5.2.2. Event Indicators by Scope and Geographic Area

All noise limits based on event indicators are grouped by geographic area and presented for emission (exhaust, tire, and engine) and outdoors in Figure 4, Figure 5, Figure 6 and Figure 7. The dot plots show the noise limits for each scope, grouped according to geographic area. Each dot represents the value of a single noise limit; the dotted lines illustrate the minimum, median, and maximum values; *n* indicates the number of noise limits represented in each figure.

Event indicators (e.g., *L_Amax_*) are used more frequently to set noise emission limits on engine (*n* = 134) and exhaust (*n* = 116) than they are for tire (*n* = 21) and outdoor (*n* = 15) noise limits.

For exhaust noise (see Figure 4), the lower values were identified in Europe, while the higher values are identified in Australia. Indeed, the maximal value identified in Europe was lower than the minimal value identified in Australia and North America. For tire noise (see Figure 5), the noise limits were only identified in Europe. For engine noise (see Figure 6), the medians were relatively close between Australia (78 dB(A)), Europe (76 dB(A)), and North America (83 dB(A)). It can be noticed that noise limits for emission scopes (exhaust and engine noise combined) were more concentrated in Europe (max – min = 15 dB(A)) than those identified in Australia (max – min = 29 dB(A)) and North America (max – min = 31 dB(A)). Only one event indicator was found for indoor noise limits: the USA [46] sets *L_A10_*_%_ = 55 dB(A) (not represented).

To conclude, it was difficult to draw meaningful conclusions due to the variability in noise limit scopes and associated acoustic indicators. In the following section, outdoor noise limits based on overall indicators were converted using Brink’s method to compare them with the WHO recommendations.

### 5.3. Comparison of Outdoor Road Noise Limits with the WHO Recommended Values

#### 5.3.1. Measurement Periods Compared to WHO Recommendations

The measurement periods varied between countries, states, and provinces as shown in Table 6. Some noise policies use *L_Aeq_* overall indicators but on specific day, evening, or night periods. For clarity and interest of concision, the noise limits defined for *L_Aeq_* indicators in noise policies for a day, evening, or night period were identified in this paper as *L_d_*, *L_e_*, and *L_n_* in Table 6.

For Australian states, the start and end times for the day and night are defined differently between New South Wales [20,21] and Western Australia [17,18]. The same applies to European countries, where measurement periods are defined differently between, for instance, Spain [32], Finland [27], France [28], and Denmark [26]. A local difference is therefore noticed in the definition of the measurement periods in the same geographic area/country (e.g., Australia and Europe).

Some measurement periods, shown by the shaded boxes in the Table 6, are not compatible with Brink’s method (see the applicable measurement periods in Table 4), specifically in New South Wales [20,21], South Australia [22], Queensland [16], but also in Finland [27]. The associated noise limits were therefore not converted. For the other noise limits, they were converted to the indicators used by the WHO’s *L_den_* and *L_n_* indicators and defined for the following periods: day—between 7 and 19 h; evening—between 19 and 23 h; night—between 23 and 7 h.

The adaptation of the measurement periods by country, state, and province is highly cultural. It is related to different ways of life, typical working hours, usual bedtimes, nightlife, and use of outdoor public spaces among other factors (see Guastavino, (2021); for further discussion on cultural differences for time periods in European noise policies, see [61]).

#### 5.3.2. Noise Limits by Zone, Compared to the WHO’s Recommendations

One hundred and eleven noise limits from Australia, Canada, USA, and Europe (both within and outside the EU) meet the conversion criteria for Brink’s method. These noise limits were converted to *L_den_* and *L_n_* (shown in Figure 8) and presented by zone: industrial, residential, and sensitive. Each dot corresponds to a converted noise limit, and the dotted lines indicate the minimum, median, and maximum values as well as the WHO’s recommendations. The noise limits presented in Figure 8 illustrate where they are in comparison with the WHO’s recommendations.

In general, noise limits identified in industrial zones were higher than those identified in residential and sensitive zones. For instance, the maximum values shown in Figure 8 (*L_den_*) were 80 dB(A), 78 dB(A), and 70 dB(A), respectively, for industrial, residential, and sensitive zones. In residential zones, the *L_den_* and *L_n_* medians were 70 dB(A) and 63 dB(A), respectively, corresponding to limits identified in California [47] and Spain [32].

With the exception of Sweden [34], all the noise limits identified in this review were higher than the 53 dB(A) (*L_den_*) and 45 dB(A) (*L_n_*) recommended by the WHO [1]. Moreover, the converted noise limits identified in sensitive zones also exceeded the WHO recommendations for residential zones, while they are more sensitive to noise than residential zones.

#### 5.3.3. Noise Limits by Type of Road

In addition to zones, noise limits may be defined also according to the type of road (i.e., existing and planning new/upgraded roads), while no specification is given on the application of the WHO’s recommendations according to the type of road. As shown in Table 7, noise limits are defined mainly for upgraded and new roads. Some countries clearly defined noise limits according to the type of road (i.e., existing, upgraded, and new roads) as in Germany [29,30,31], New South Wales [21], and Queensland [16]. Other countries defined noise limits according to only one type of road (e.g., Denmark [26], Sweden [35], Andalusia [33], and Spain [32]). Noise limits defined for existing roads were generally less restrictive than those defined for upgraded and new roads, because mitigation measures can be implemented during road improvement and maintenance works. Another approach was proposed in Denmark [26]: the noise limits were dedicated to planning/existing dwellings near existing roads. However, it was also recommended to consider the noise impact and secure the lowest possible noise level when planning new/upgraded roads [26].

#### 5.3.4. Measurement Protocols for Outdoors

Noise policies set protocols to ensure that measurements were as representative as possible of the noise environment and were minimally affected by environmental factors (e.g., reflective surfaces close to the microphone, weather conditions), which may cause noise or increase the measured sound pressure level. Table 8 summarizes the different measurement protocols for each country, province, and state including the type of sound level meter, the calibration, and the acoustic standards.

The noise limits are mainly defined at the facade of noise-sensitive premises, but they can be set at the property lines (e.g., Alberta [39] and Illinois [46]) and roadside (e.g., Quebec [42]). The sound level meter is most often set at 1.5 m above the ground but can also be set at 1.2 m, 1.4 m, and 1.8 m. The Queensland [16] protocols also require a height of 4.6 m above the ground in order to measure the noise impact to the second floor of noise-sensitive buildings. Regarding the distance between the microphone of the sound level meter and the reflective surfaces, Australian protocols differ from American and European ones. In the USA and Europe, the microphone is set at a distance between 2 or 3 m from reflective surfaces to avoid an increase in the sound level due to the fact of acoustic reflections. In Australia, the microphone may be set at a distance of 1 m from reflective surfaces (other than the ground) to account for facade reflections. Moreover, before the construction of new dwellings, measurements are taken in free field with a penalty of 2.5 dB(A) added to the measured noise level to account for noise reflections of future facades of dwellings. Thus, unlike the USA and Europe, Australian noise limits factor facade reflections into their measurements.

Concerning weather conditions, which may also bias the measurement, some policies set specifications. For instance, the wind speed should not exceed 11 km/h in Queensland [16] and Western Australia [17,18] or 20 km/h (with the use of windscreen) in Illinois [48] and California [47]; the temperature must be between −10 and 50 °C; either the rainfall does not exceed 0.3 mm/h or the road surface must be dry. In addition, the weather conditions must be recorded during measurements in Queensland [16] and Western Australia [17,18].

The referred standards for a sound-level meter are AS 2702, AS IEC 61672, NF S 31-110, and ANSI S1.4. In Western Australia [17,18], Illinois [48], and California [47] refer to the mentioned standards for the sound-level meters of Type I and Type II, indicating the degree of precision for the sound-level measurements. Western Australia [17,18] refers also to the AS IEC 60942 standard for the calibration of a sound level meter. Generally, the sound-level meter is calibrated before and checked after each measurement. The sound-level meter must also be calibrated in laboratory and at least once every two years in Western Australia [17,18]. The calibration of the sound-level meter ensures repeatability on the measurements in order to avoid any bias on the measured sound level. In addition, the following acoustic standards, AS 2702 and NF S 31-085, may also refer to measurement protocols. These standards provide measurements specifications in terms of sound level meter type, calibration, measurement location, etc., to ensure that measurements are systematically conducted, independently of the operator.

## 6. Discussion

In this section, the influences of centralized and decentralized approaches on the noise policies definition are presented. Afterwards, the disparities between noise policies and WHO recommendations regarding acoustic indicators, measurement periods, noise limits, zones and measurement protocols are discussed.

### 6.1. Centralized and Decentralized Approaches

The detrimental effects of road traffic noise on human health can be prevented to some extent through the implementation of guidelines, directives, policies, and regulations. Furthermore, they can be set at different levels of administrative government and target either different scopes through limits of noise emission or limits of exposure.

The EU uses a centralized approach (END [11]) to harmonize noise maps (including road noise) between Member States by proposing a directive and a common methodology (CNOSSOS-EU [12]). D’Alessandro and Schiavoni [10] showed that the use of a common approach among Member States of the EU could improve noise efforts, provided that Member States adopt entirely the END into their own noise policies.

For emission scope, we showed that the centralized approach is mainly used for vehicles before commercial release at the highest administrative government of a country or a geographic area. For instance, in Europe, common regulations are proposed for all EU Member States. By agreeing on a common vehicle noise emissions standard, the EU has set the most ambitious targets for new vehicles. One potential reason for this is the larger vehicle market share in EU in comparison with smaller countries. Regarding noise limits for emission scope, there is a greater decibel range in Australian and North America noise limits for noise emissions regulations (i.e., engine and exhaust combined) than those set by the EU (see Figure 4 and Figure 6). It could be explained by the fact that noise regulations are set by the EU, providing a general direction for all EU Member States. In comparison, vehicle noise emission regulations are mainly regulated at the state level in Australia and North America. Each state can therefore propose different noise limits in the absence of a shared national approach.

Moreover, the differences between noise limit values identified in Australia, Europe, and North America can be explained by the different measurement protocols described in their noise regulations same as the engine speed (for exhaust noise), the vehicle speed (for engine noise), the vehicle categories, and the microphone location (see Table A1 in the Appendix A). For instance, the engine speed during exhaust measurements reported in Australia [19,52] can exceed the speeds used in Europe [23,24], while the vehicle passage speed during engine measurements can be higher in USA [45] than in Australia [19,52] and Europe [23,24]. Also, noise limits are defined for several categories of vehicle, dedicated either to freight transport or passenger transport (four wheels or less than four wheels). Finally, the microphone locations were closer to vehicles in Australia [19,52] and Europe [23,24] than in the USA [45].

For outdoor and indoor scopes, noise policies were decentralized because they were set at the provincial/state levels in Australia and North America and at the national level in the EU. We showed in Section 5.3.1. that a local difference may be noticed in the same geographic area/country (e.g., definition of measurement periods). A decentralized approach allows for greater flexibility and adaptation to local contexts, such as working hours, involving rush hours and, therefore, noisy hours, or on the contrary, bedtime, involving times for which noise should have the least possible impact. On the other hand, the decentralized approach for outdoor and indoor may be noticed by the dispersion of noise limit values especially in Europe and North America (see Figure 2 and Figure 3). However, the values were less spread out across Australian states (decentralized approach) than in Europe and North America, while Burgess and Macpherson noticed that there was little common ground in noise limits among Australian states [4].

Our findings indicate that measurement protocols varied significantly among noise policies across geographic areas. For instance, Australian noise limits were defined by taking into account the facade reflections, but this is not considered in the USA and Europe. This means that Australian noise limit factors facade reflections into their measurements. It is also shown in Table 8 that noise policies refer to acoustic standard in the definition of measurement protocol. However, countries and states do not rely on the same acoustic standard, which explains the differences encountered between measurement protocols identified in noise policies. In contrast, similarities are identified between state and provincial noise policies within the same country. For instance, the ANSI S1.4 acoustic standard is referred for the type of sound level meter in Illinois [48,49] and California [47]. The same applies for AS 2702 acoustic standard referred for measurement protocols between Queensland [16] and Western Australia [17,18]. The distance of 1 or 3.5 m (by adding 2.5 dB to the measured noise level) from reflecting surfaces are also the same in Queensland, New South Wales [21], South Australia [22], and Western Australia, while the heights of the microphone above the ground are different between Queensland (1.8 m), New South Wales (1.2 m), and Western Australia (1.4 m). The same weather conditions are also referred to in Illinois and California such as a wind speed of 20 km/h (with windscreen), a temperature range of −10 and 50 °C, and the road surface must be dry during acoustic measurements. In Queensland and Western Australia, the same wind speed of 11 km/h can be found in their respective noise policies.

Finally, harmonizing noise management strategies across policies could provide a common approach throughout the world, which would allow to assess and compare different populations’ noise exposure (an observation already noted for Member States of the EU [9]) and to adopt unified noise mitigation efforts [10].

### 6.2. Disparities between Noise Limits from Noise Policies Compared to the WHO Recommendations

Brink’s method for converting noise limits into common acoustic indicators allows for a comparison of outdoor noise limits with the WHO’s recommendations. Some limits are defined by Brink et al. [8] for the application of their conversion factors: the vehicle types and driving behaviors should be similar to Western European countries. However, we assumed in the Section 4 that the selected Western countries were similar to Western European’s countries in terms of vehicle types (i.e., noise emissions) and driving behaviors, while we showed that there are many differences between noise limits for emission regulations (engine and exhaust) across the three geographic areas, leading to differences in vehicle noise emissions. Moreover, driving behaviors are part of traffic culture, which is unique to each country [62]. Therefore, Brink’s method may be less suited to the study of countries where these factors might differ from Westernern Europe. In addition, this method is applicable only to overall indicators and specific measurement periods, while we showed that different event indicators and other measurement periods were also used extensively in noise policies across different countries.

In addition, the WHO’s recommendations are based exclusively on overall indicators, *L_den_* and *L_n_*, but some countries use event indicators (e.g., *L_Amax_*, *L_A10%_*, *L_A5%_*) to characterize noise events in order to complete overall indicators (e.g., *L_Aeq_* and derivates). According to the WHO, the use of event indicators, such as *L_Amax_*, is warranted in specific situations, such as nighttime, where noise events can disrupt sleep and lead to other physiological reactions with longer-term implications. Nonetheless, the latest WHO recommendations (2018) dedicated to outdoors and road noise does not specific limit values using event indicators, because the scientific evidence on the relationship between event indicators and long-term health outcomes is limited [1]. For night noise, but without specifying the noise source, the WHO (2009) states, however, that there is sufficient evidence to recommend indoor values based on *L_Amax_*, varying from 32 dB(A) to 42 dB(A), in order to avoid biological effects due the fact of noise exposure and to ensure quality of sleep [63].

Local and cultural adaptations by administrative governments is shown in Section 5.3.1. for the setting of the times of the day (corresponding to measurement periods), even in the same geographic area/country such as in Australian states and Member States of the EU (see Table 6). This local difference was not taken into account in the WHO’s recommendations, because the day, evening, and night were defined for specific measurement periods on the *L_den_* and *L_n_* indicators. The WHO defined measurement periods which did not account for the local context—people’s schedules and activities (e.g., working hours and usual bedtimes). Therefore, countries, states, and provinces should adopt the values recommended by the WHO but using definitions of measurement periods for day, evening, and night that are locally and culturally meaningful and appropriate.

Noise limits are defined according to zones, which we categorized as sensitive, residential, mixed, and industrial. Our findings indicate that all noise limits identified in residential zones (including mixed and undefined zones) exceeded the noise limit recommendations provided by the WHO, by a median of 17 dB(A) (*L_den_*) and 18 dB(A) (*L_n_*). In 2019, a similar study conducted by Peeters and Nusselders [6] showed that the majority of noise limits also exceeded the WHO’s recommendations. However, their study was limited to noise policies from EU Member States and without distinction between noise limits based on *L_day_* and *L_den_*. Moreover, we showed that noise limits in sensitive zones also exceeded the WHO’s recommendations, despite a higher sensitivity to noise and the greater potential for health-related effects due to the fact of noise. However, the only noise policy identified after 2018 (when the latest WHO recommendations were published) was the Western Australia policy (2019) [18,19], the other noise policies presented in this paper should therefore be updated so that the requirements tend towards the latest WHO recommendations.

Interim target values can be defined for countries where the latest WHO recommendations cannot be achieved in the short term. The concept of an interim target was proposed by the WHO (2009) for night noise (without specifying the noise source) for policy-makers who choose a stepwise approach [63]. The outdoor interim target, defined by the WHO (2009), is 15 dB(A) higher than the recommended *L_n_* = 40 dB(A), also defined by the WHO (2009). Which corresponds to within a 2 dB(A) (*L_den_*) and a 3 dB(A) (*L_n_*) gap between the WHO’s recommendations (2018) and the medians identified for outdoor noise limits in residential zones. The EU also uses interim targets for emission noise by setting progressively more restrictive noise limits, with small gaps between target and interim targets of maximum 2 dB(A), over a ten-year period entering into effect in 2016, 2022, and 2026 [24]. However, to reach the WHO’s recommendations, it is important to consider strategies that go beyond noise mitigation and to include urban planning practices by integrating noise considerations throughout the planning process [9].

Finally, the WHO’s noise guidelines are recommended for all roads, whether they are existing, upgraded, or new. However, noise mitigation (e.g., noise barriers on highways) could be implemented during road improvement and construction works to minimize noise propagation in order to reach the WHO’s recommendations. This is also the reason why the noise limits identified in policies and defined according to the type of road may be more restrictive for upgraded and new road. On the other hand, the British Columbia [40] policy defines different thresholds (according to the emergence of the noise level after a road project): no impact on the communities, minor, moderate, and severe. If the project severely or moderately impacts communities, it shall be warranted to implement noise mitigation measures. More generally, population growth puts pressure on municipalities to expand, leading to the construction of new residential zones, sometimes too close to existing roads. Noise limits are therefore also set when planning dwelling areas near existing roads such as in Denmark [26]. Considering noise proactively in the early stages of planning could prevent noise issues by prohibiting or limiting new dwellings in the vicinity of area exposed to high noise levels.

## 7. Conclusions

This study provides support for the importance of a centralized approach to regulating noise, particularly when dealing with noise emissions from vehicles, noise maps, and action plans. That said, some noise policies require a more decentralized approach to adapt to their local contexts. We show that a centralized approach makes it possible to better control noise in a territory using a coherent and common approach between lowest administrative governments of a geographic area or a country. While a decentralized approach allows noise policies to be adapted to the local contexts of a province or state (e.g., Australia and North America) or even countries (e.g., EU Member States). However, decentralized approaches can lead to large variations on noise limits and differences in measurement protocols in the same geographical area (e.g., Australia, North America, and EU).

This study also suggests the use of multiple indicators to address the complexity of road noise, such as overall indicators (e.g., *L_den_* and *L_n_*) and event indicators (*L_A10%_* and *L_Amax_* during nighttime and *L_A5%_* for rush hours), despite the fact that the WHO (2018) recommendations were defined only using overall indicators. Moreover, the WHO (2018) recommendations were defined on specific times of the day; they did not take into account the local specificities of each country such as working hours (involving rush hours and, therefore, noisy hours) or, on the contrary, typical bedtimes (involving the need for quiet). Therefore, noise policies should implement the latest WHO recommendations using their own definition of times of day in order to account for their local specificities. Then, the noise limits identified from all zones exceeded the WHO’s recommendations. Noise policies should therefore be updated so that the requirements tend towards the latest WHO recommendations using interim target values when WHO recommendations cannot be achieved in the short term. In addition, if stricter noise limits are set for new and upgraded roads, noise abatement measures and other noise management strategies could be implemented at the same time to minimize noise propagation and converge towards the WHO’s recommendations. Lastly, noise limits could be defined when planning for dwelling areas near existing roads in order to prohibit building new dwellings too close to roads. It is therefore critical to integrate noise considerations into urban design and planning (e.g., noise abatement whether for road or dwelling) and to do so from the early stages and throughout the entire process. Future directions include extending this analysis to other countries and other sound sources, identifying international best practices for environmental noise management, and informing innovative noise policies.

## Figures and Tables

**Figure 1 ijerph-19-00173-f001:**
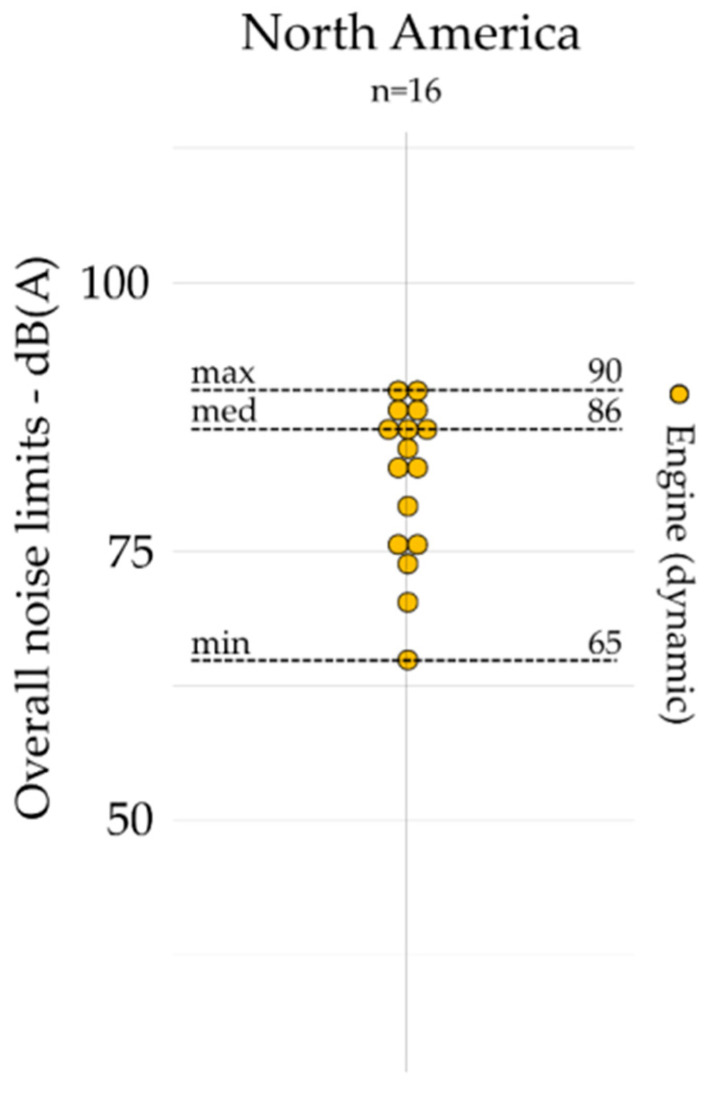
Overall indicators for emission scope (i.e., engine) identified in North America (*n* = 16).

**Figure 2 ijerph-19-00173-f002:**
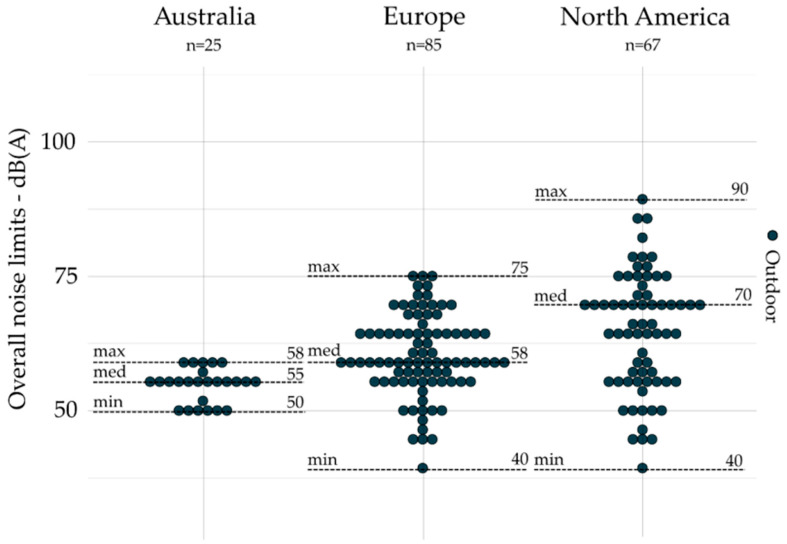
Overall indicators (without conversion by Brink’s method) for outdoors (all zones combined) grouped by geographic area (*n* = 177).

**Figure 3 ijerph-19-00173-f003:**
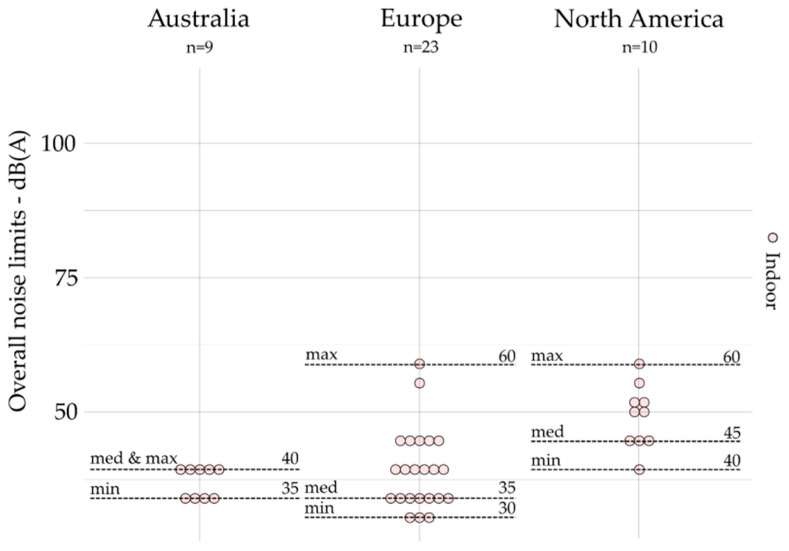
Overall indicators for indoors (all zones combined) grouped by geographic area (*n* = 42).

**Figure 4 ijerph-19-00173-f004:**
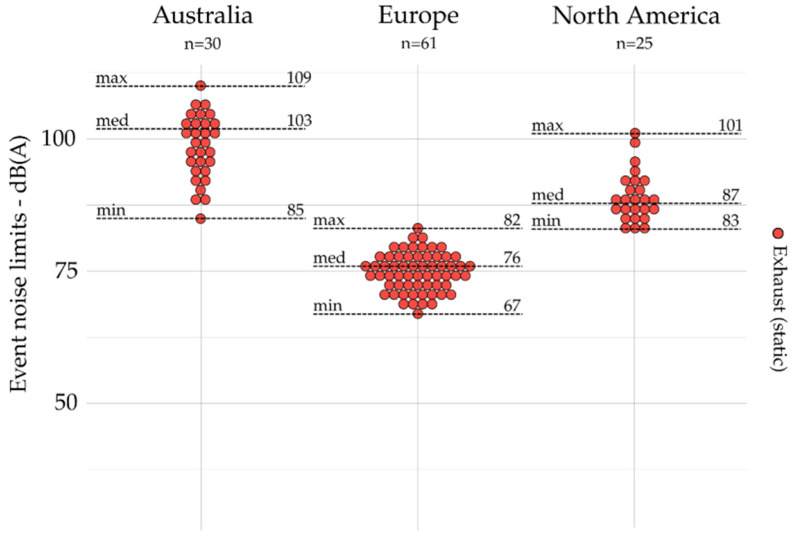
Event indicators for emission (i.e., exhaust) and grouped by geographic area (*n* = 116).

**Figure 5 ijerph-19-00173-f005:**
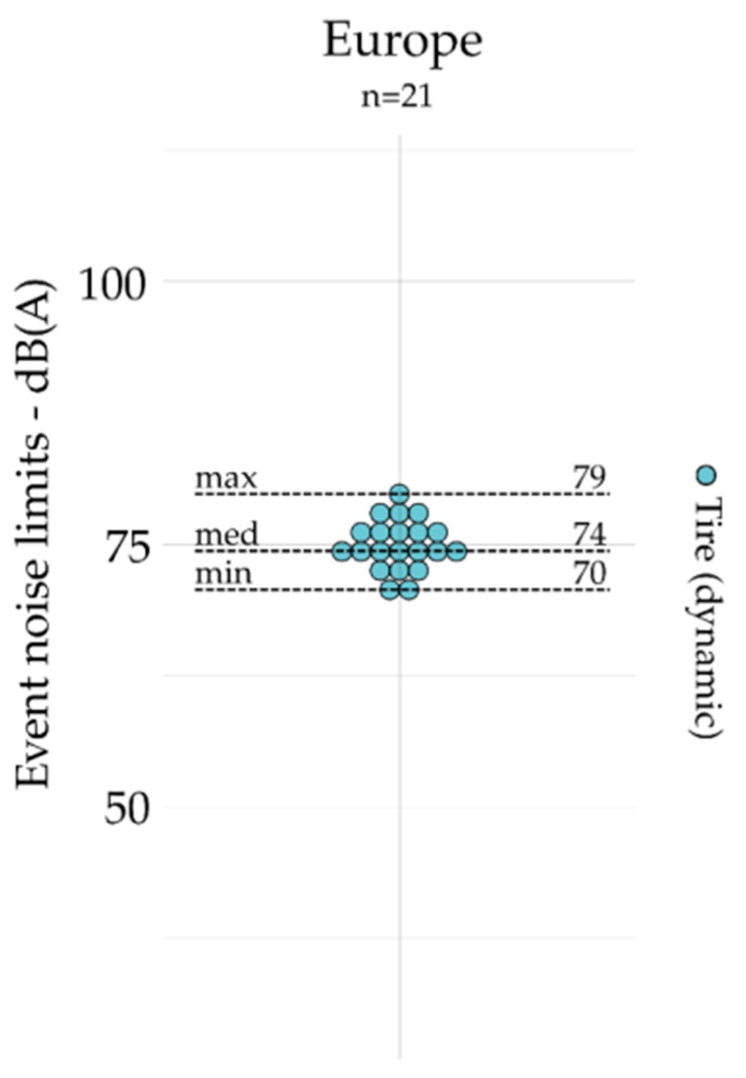
Event indicators for emission (i.e., tire) and grouped by geographic area (*n* = 21).

**Figure 6 ijerph-19-00173-f006:**
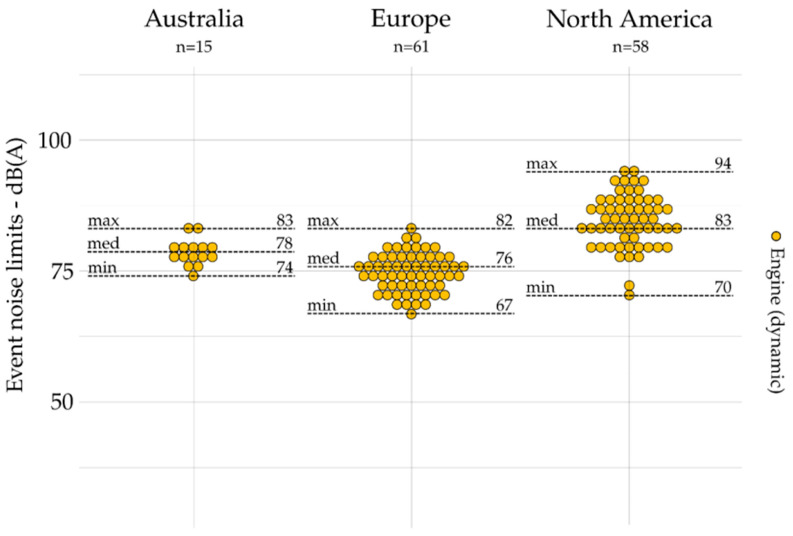
Event indicators for emission (i.e., engine) and grouped by geographic area (*n* = 134).

**Figure 7 ijerph-19-00173-f007:**
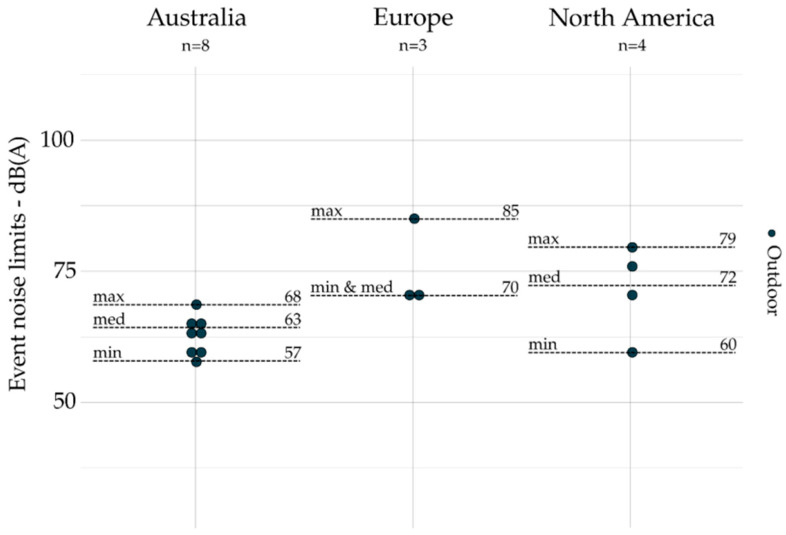
Event indicators for outdoors (all zones combined) and grouped by geographic area (*n* = 15).

**Figure 8 ijerph-19-00173-f008:**
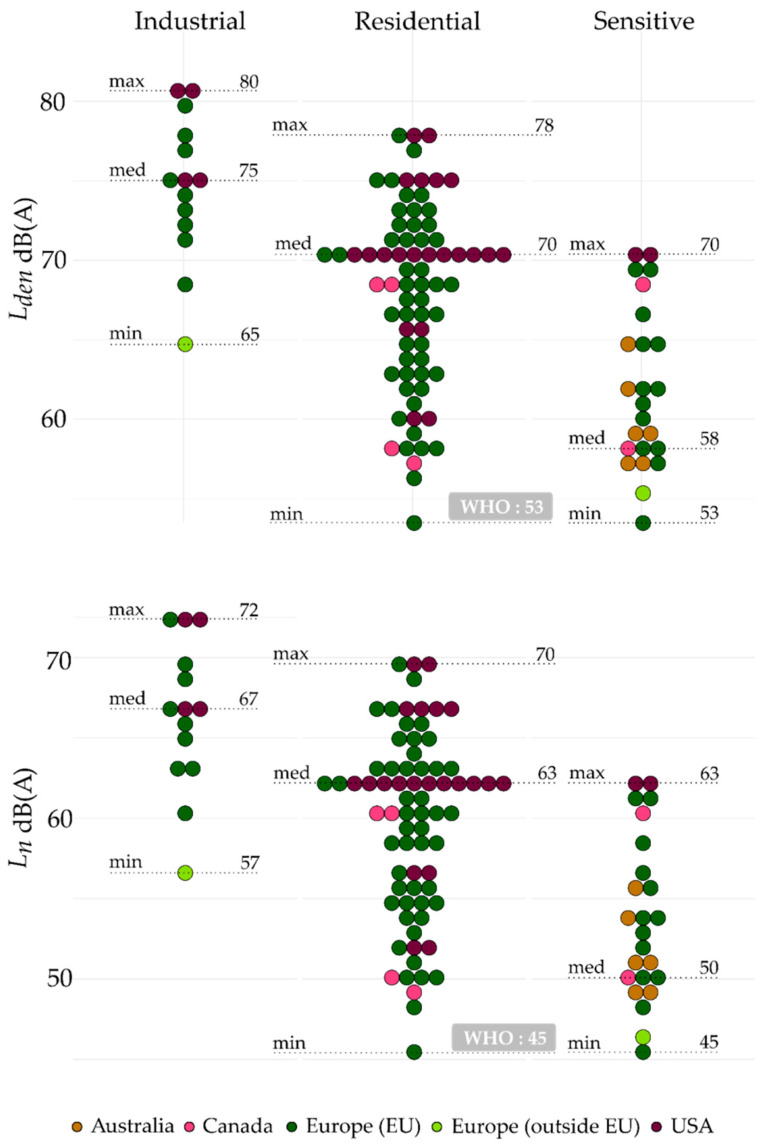
Outdoor noise limits converted to *L_den_* (upper plot) and *L_n_* (lower plot) by Brink’s method and grouped by zone (industrial, residential, and sensitive). “WHO” indicates the limit recommended by the World Health Organization.

**Table 1 ijerph-19-00173-t001:** Definition of acoustic indicators identified in noise policies and regulations (“A” stands for A-weighted dB scale).

**Overall Indicators**	
*L_Aeq_*_,*T*_, *L_d_*, *L_e_*, *L_n_*	Equivalent sound level over a T period (hour, day, evening, night, etc.)
*L_dn_*, *L_den_*	Equivalent sound level over a 24 h period with a penalty added for noise during the nighttime or during the evening and nighttime hours, respectively. The day *d*, evening *e*, and night *n* are defined at different times depending on the country (see Section 5.3.1).
*L_Ar_* _,*T*_	Equivalent sound rating level over a T period, adjusted according to the nature of the noise (e.g., tonal, impulsive, low frequency)
**Event indicators**	
*L_Amax_* _,*T*_	Maximum sound level over a T period, measurement with time-constant (e.g., slow (=1 s), fast (=0.125 s))
*L_A5%_*_,*T*_, *L_A10_*_%,*T*_	Sound level exceeded for 5% or 10% over a T period, calculated by statistical analysis

**Table 2 ijerph-19-00173-t002:** Four categories of zones used to map noise limits.

Name	Description
Sensitive	Zones which have a primary use that is noise-sensitive or requires special attention (e.g., health and educational premises, quiet zones).
Residential	Zones destined for residential buildings, regardless of the density.
Mixed	Includes a mix of residential, commercial, and office spaces as well as industrial spaces that generate a moderate noise level.
Industrial	Zones destined for industrial activity, regardless of the intensity of the activity.

**Table 3 ijerph-19-00173-t003:** Definition of acoustic indicators identified in noise policies and regulations.

Geographic Area—Supranational, National, State/Provincial		Emission (Engine, Exhaust, Tire Noise)	Outdoors	Outdoors (Fulfilling the Brink Criteria)	Indoors
Australia	Australia [15]	X			
	Queensland [16]		X	X	
	Western Australia [17,18,19]	X	X	X	X
	New South Wales [20,21]		X	X	
	South Australia [22]		X		
Europe	EU [23,24,25]	X			
	EU—Denmark [26]		X		
	EU—Finland [27]		X	X	
	EU—France [28]		X	X	X
	EU—Germany [29,30,31]		X		X
	EU—Spain [32]		X	X	X
	EU—Andalusia [33]		X		X
	EU—Sweden [34,35]		X		X
	Outside EU—Norway [36]		X		X
	Outside EU—Switzerland [37]			X	
North America	Canada [38]	X			
	Canada—Alberta [39]		X		X
	Canada—British Columbia [40,41]	X	X		
	Canada—Quebec [42,43]	X	X		X
	Canada—Ontario [44]		X	X	X
	USA [45,46]	X	X	X	
	USA—California [47]		X		X
	USA—Illinois [48,49]	X	X	X	
	USA—Washington [50,51]	X			
Noise Limits (total)		287	192	111	43

**Table 4 ijerph-19-00173-t004:** Applicable measurement periods for Brink’s method.

Day	Evening	Night
*L_d_*	*L_e_*	*L_n_*
6 h–22 h 7 h–23 h 7 h–19 h	18 h–22 h 19 h–23 h	22 h–6 h 23 h–7 h
*L_dn_*		
6 h–22 h 7 h–22 h 7 h–23 h		22 h–6 h 22 h–7 h 23 h–7 h
*L_den_*		
6 h–18 h 7 h–19 h	18 h–22 h 19 h–23 h	22 h–6 h 23 h–7 h

**Table 5 ijerph-19-00173-t005:** Overall and event indicators used by countries, states, and provinces in noise policies and regulations.

Acoustic Indicators	Countries, States, and Provinces
Overall Indicators	
*L_Aeq_*	Australia [22], Finland [27], France [28], Germany [29,30,31], Sweden [34,35], Queensland [16], Western Australia [17,18], New South Wales [20,21], South Alberta [39], Quebec [42,43], Ontario [44], Illinois [48,49].
*L_d_*, *L_e_*, *L_n_*	Andalusia [33], Spain [32]
*L_dn_*	British Columbia [40], California [47]
*L_den_*	Denmark [26], Norway [36]
*L_Ar_*	Switzerland [37]
**Event Indicators** (maximal and statistical)	
*L_Amax_*	Australia [15], British Columbia [40,41], Canada [38], EU [23,24,25], Quebec [43], USA [45], Sweden [35], Washington [50,51], Western Australia [19]
*L_A5%_*	Norway [36]
*L_A10%_*	USA [46], Queensland [16]

**Table 6 ijerph-19-00173-t006:** Day, evening, and nighttime measurement periods for each geographic area. The shaded boxes represent the measurement periods not compatible with Brink’s method (see Section 4.4.).

Countries, States, and Provinces	Acoustic Indicators	Day	Evening	Night
Andalusia [33], Spain [32]	*L_d_*, *L_e_*, *L_n_*	12 h (7–19 h)	4h (19–23 h)	8 h (23–7 h)
Finland [27], New South Wales [20,21], South Australia [22]	*L_d_*, *L_n_*	15 h (7–22 h)		9 h (22–7 h)
Ontario [44], WHO [1]		16 h (7–23 h)		8 h (23–7h)
France [28], Germany [29,30,31], Western Australia [17,18]		16 h (6–22 h)		8 h (22–6 h)
Queensland [16]	*L_d_*	18 h (6–22 h)		
		12 h (6–18 h)		
Alberta [39], Quebec [42], Sweden [34,35]	*L_Aeq_* _,*24h*_	24 h		
British Columbia [40], California [47]	*L_dn_*	24 h (7–22 h/22–7 h)		
Denmark [26], Norway [36], WHO [1]	*L_den_*	24 h (7–19 h/19–22 h/22–7 h)		

**Table 7 ijerph-19-00173-t007:** Noise limits defined according to the type of road.

Countries, States and Provinces	Type of Road		
	Existing	Upgraded	New
France [28], South Australia [22], Western Australia [17], Alberta [39], British Columbia [40]		X	X
Andalusia [33], Spain [32]			X
Denmark [26], Sweden [35]	X		
Germany [29,30,31], New South Wales [21], Queensland [16]	X	X	X

**Table 8 ijerph-19-00173-t008:** Specifications of the measurement protocols in noise policies.

	Countries, States, and Provinces	Specifications
Acoustic standards	Sound level meter	
	Queensland [16]	AS 2702 ^5^
	Western Australia [17,18]	AS IEC 61672 ^6^ (type I and II)
	France [28]	NF S 31-110 ^7^
	Illinois [48], California [47]	ANSI S1.4 ^8^ (type I and II)
	Calibration	
	Western Australia [17,18]	AS IEC 60942 ^9^
	Measurement protocols	
	France [28]	NF S 31-085 ^10^
	Queensland [16], Western Australia [17,18]	AS 2702 ^11^
Calibration	Queensland [16], Western Australia [17,18], California [47], Illinois [46]	before and after each measurement
	Queensland [16], Western Australia [17,18], California [47], Illinois [46]	laboratory
Measurement location	Height of the microphone above the ground	
	New South Wales [21], South Australia [22], Ontario [44], Illinois [48], California [47]	1.5 m
	Alberta [39]	1.2 m
	Western Australia [17,18]	1.4 m
	Queensland [16]	1.8–4.6 m
	Distance from reflecting surfaces (facade or property line)	
	Queensland [16], Western Australia [17,18], New South Wales [21], South Australia [22]	1 or 3.5 m (+2.5 dB)
	Alberta [39], France [28]	2 m
	Ontario [44]	3 m
Weather conditions	Maximal wind speed	
	Queensland [16], Western Australia [17,18]	11 km/h
	Illinois [48], California [47]	20 km/h (with windscreen)
	Temperature	
	Illinois [48], California [47]	−10 and 50 °C
	Rainfall	
	Queensland [16]	rainfall 0.3 mm/h
	Western Australia [17,18], Illinois [48], California [47]	road surface must be dry

^5^ Methods for the measurement of road traffic noise; ^6^ electroacoustics–sound-level meters; ^7^ Characterization and measurement of environmental noise; ^8^ specifications for sound-level meters; ^9^ electroacoustics–sound calibrators; ^10^ characterization and measurement of noise due to the road traffic; ^11^ methods for the measurement of road traffic noise.

## Data Availability

The data supporting reported results will be available soon on McGill University Dataverse. The identified values for all sources (including road noise) are gathered in “TableValue_Cleaned_Final.xlsx”. The programming code and the associated table allowing to plot Figure 1, Figure 2, Figure 3, Figure 4, Figure 5, Figure 6 and Figure 7 are respectively, “All identified values by continent_indicators_location.R” and “All identified values by continent_indicators_location.csv”. The programming code and the associated table allowing to plot Figure 8 are respectively “Brink_outdoor_Ln_Lden_zoning_ind_resmixnone_sensible.R” and “Brink_outdoor_Ln_Lden_zoning_ind_resmixnone_sensible.csv”. Additional details are given in programming codes in comments.

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
