# Peer review of "Comparison of Road Noise Policies across Australia, Europe, and North America"

_ijerph, 2021, doi:10.3390/ijerph19010173_

Round 1
Reviewer 1 Report
Overall, I thought this paper comparing road noise policies across Australia, Europe and North America was a worthy undertaking.
Do reread for some minor grammatical corrections, e.g 1) In the abstract: This is particularly critical...identified as one of the main culprits....2) 1st paragraph in body of paper: Among the sources...contributing to. Just reread and you will be able to correct accordingly.
Stronger statements on adverse impacts of road noise on mental and physical health are needed to stress how important it is to move more assertively to reduce road noise. Some people consider aircraft noise to have a greater adverse impact and this should be addressed.
Recommendations that planners and designers should pay more attention to noise abatement re: road design should be elaborated.
Author Response
Dear reviewer,
The authors would like to thank you for your thorough reviews and comments. They were incorporated into the revised version of the manuscript by using the “Track Changes” in Word. For ease of reading, the modified text passages are inserted into the attached document, with identified line numbers of the manuscript (highlighted in yellow). The identified line numbers correspond to the version of the manuscript in “Track Changes”. We hope that the revised version addresses your concerns.
Best regards.

Reviewer 2 Report
General comments
The paper is interesting, it deals with an important topic. It has a very ambitious goal which is two-fold: on one hand there is a comprehensive review of noise indicators that are used across noise policies internationally. On the other hand, there is a methodological approach or at least attempt at harmonization.
The choice about noise policies from Australia, Europe and North America because of their similarities to the Canadian context sounds a bit shaky to me. For instance, one could argue that Canada is quite different in terms of densities, agglomerations and land use more generally from Europe, thus noise mapping technique may need to take into account different factors and corrections.
Similarly, to the point above, even if this is out of the scope of the current review, I think it would be interesting to mention (even if only briefly) what the regulatory frameworks and noise indicators look like in other regions of the world? I am thinking about Asia, Africa, Latin America…
Section 4.3 – I am not totally clear about how these 522 noise limits were identified? How were they retrieved – could the search strategy be discussed more in detail for the sake of replicability?
Minor comment
In Table 1 in the first row, I would probably say for LAeq,T: (e.g., hour, day, evening night) – having day, evening, night only seems to limit T to the Lden intervals?
Author Response

(The authors gave the same response as above.)

Reviewer 3 Report
- Table 1: second row: please be more precise about the T period for L_{Aeq,T} (what are the usual values for T?), L_d, L_e, and L_n (between which and which hours?). Also, in second and fourth and sixth rows, please mention that the "A" stands for A-weighted dB scale. Maybe these indications can simply be indicated in the caption of the figure, it would therefore nicely complement the main text.
- L145-146: the "cultural similarity to Quebec" is unclear... what would it be? Is it in terms of language, economics, leisure, etc.?
- L146: in a similar way, the "similarity of climatic conditions with Quebec" is to be explained. One could argue that, e.g., Spain (table 3) offers a quite different climate from Quebec.
- Section 4.4: the definition of "day", "evening", and "night" is partly cultural, and is related to people's schedule and activities, bedtime, etc. This might explain some differences between measurements and regulations, and be worth being taken into account when trying to find a common measure. It think we agree with the authors on this point, but I think this point might be emphasized. Maybe it should already be discussed here (or later near L370), introducing and expanding the short discussion in L576-584.
- Figures 1, 2, 3, and L311: I guess n is the number of indicators the authors found, but it should be explained (maybe around L299-300).
- Figure 8: It took me some time to make sure the upper panel presents L_{den} and the lower panel presents L_n. I would suggest moving the label of y-axis to the left-hand side of both panels, and include an explanation within the caption of the figure.
Suggestions and typo:
- L37: contribute -> contributing
- Table 3: inconsistency between the "-" signs separating national/supranational from state/provincial levels.
- Please double-check grammar in the sentence in L582-583, as I'm not sure to understand what is meant.
Author Response

(The authors gave the same response as above.)
